# Development of Autonomous Electric USV for Water Quality Detection

**DOI:** 10.3390/s25123747

**Published:** 2025-06-15

**Authors:** Chiung-Hsing Chen, Yi-Jie Shang, Yi-Chen Wu, Yu-Chen Lin

**Affiliations:** 1Department of Telecommunication Engineering, National Kaohsiung University of Science and Technology, Kaohsiung 811213, Taiwan; chiung@nkust.edu.tw (C.-H.C.); 72325@nkust.edu.tw (Y.-C.W.); c108185117@nkust.edu.tw (Y.-C.L.); 2Energy and Agile System Department, Metal Industries Research & Development Centre, Kaohsiung 852005, Taiwan

**Keywords:** unmanned surface vehicle, water quality detection, automatic identification system, precise positioning

## Abstract

With the rise of industry, river pollution has become increasingly severe. Countries worldwide now face the challenge of effectively and promptly detecting river pollution. Traditional river detection methods rely on manual sampling and subsequent data analysis at various sampling sites, requiring significant time and labor costs. This article proposes using an electric unmanned surface vehicle (USV) to replace manual river and lake water quality detection, utilizing a 2.4 G high-power wireless data transmission system, an M9N GPS antenna, and an automatic identification system (AIS) to achieve remote and unmanned control. The USV is capable of autonomously navigating along pre-defined routes and conducting water quality measurements without human intervention. The water quality detection system includes sensors for pH, dissolved oxygen (DO), electrical conductivity (EC), and oxidation-reduction potential (ORP). This design uses a modular structure, it is easy to maintain, and it supports long-range wireless communication. These features help to reduce operational and maintenance costs in the long term. The data produced using this method effectively reflect the current state of river water quality and indicate whether pollution is present. Through practical testing, this article demonstrates that the USV can perform precise positioning while utilizing AIS to identify potential surrounding collision risks for the remote planning of water quality detection sailing routes. This autonomous approach enhances the efficiency of water sampling in rivers and lakes and significantly reduces labor requirements. At the same time, this contributes to the achievement of the United Nations Sustainable Development Goals (SDG 14), “Life Below Water”.

## 1. Introduction

River pollution has become a prominent environmental issue in recent years. In 2015, the United Nations formally identified water resources as a priority in the Sustainable Development Goals (SDGs) [1]. River pollution incidents frequently indicate that contamination reduces biodiversity and destroys habitats, while also causing health risks to humans who drink contaminated river water [2,3,4]. This highlights the urgent need for a rapid and accurate data collection method to promptly detect and address river water quality issues. Currently, most water quality detection methods primarily rely on personnel who regularly sample river water and send it to laboratories for analysis [5], as shown in Figure 1. Although this approach provides precise water quality data, it is labor intensive and cannot provide real-time information. Moreover, some unscrupulous individuals may discharge wastewater during the sampling intervals, making the continuous and effective monitoring of river water quality a significant challenge.

Currently, the method used for river pollution water quality assessment is the Water Quality Index (WQI). This method primarily quantifies multiple water quality parameters into a single index, which is then used to determine the pollution level or quality grade of the water body [6]. Typically, parameters such as pH, dissolved oxygen (DO), suspended solids (SS), and the biochemical oxygen demand (BOD) are measured during the assessment. The weights of these parameters are shown in Table 1. The final WQI value is calculated using a legal formula to correspond to the degree of pollution in the river [7].

In recent years, unmanned surface vehicles (USVs) have been increasingly adopted for water quality monitoring due to their autonomy, safety, and flexibility. Several international studies have demonstrated the practicality of such systems. For instance, an American research team developed a USV equipped with temperature, turbidity, and pH sensors integrated with a remote control platform to achieve real-time water monitoring with satisfactory performance in small-scale aquatic environments [8]. Similarly, Yong, X. et al. proposed an intelligent trajectory control method for water quality inspection USVs; it incorporates a human simulated intelligent control algorithm and a line-of-sight algorithm [9]. Their article confirmed that a combination of intelligent control strategies could significantly improve tracking accuracy and overall system robustness through simulation and field experiments.

Polluted rivers can be identified by detecting the pH, DO, ORP, and EC [10,11]. This article aims to develop an electric USV that can be remotely controlled and monitored for river water quality. Through remote control, the USV can conduct water quality measurements at designated locations within the river, thereby reducing the need for on-site personnel. Compared to traditional fuel engines, the pure electric propulsion system used in this USV significantly reduces noise and underwater vibrations, thereby minimizing disturbances to aquatic ecosystems. Moreover, the absence of exhaust emissions aligns with global carbon reduction initiatives and supports the sustainable development goals [12,13,14].

## 2. System Structure and Design

To realize the electric USV for remote river water quality monitoring, this article explores three main systems: a pure electric propulsion system, an autonomous navigation system, and a water quality detection system.

1.Pure electric propulsion system:

This system utilizes two DC 12 V 100 Ah lead–acid batteries (Tuflong, Japan), a 960A bidirectional brushed electronic speed controller (Holybro, SDMODEL, Taiwan), and a 300 W underwater thruster (Inphic, Taiwan) as its power source. The hull is designed to ensure sufficient ballast weight to maintain the stability of the USV and reduce the risk of capsizing due to wind, waves, or other environmental disturbances. Traditional boats using fuel engines often achieve this by introducing ballast water, which can unintentionally draw in aquatic organisms and disrupt the ecosystem. The lead–acid battery used in this system not only provides a reliable and stable energy source but also serves as an adequate ballast to ensure the boat’s center of gravity. The underwater thruster has been modified to enhance waterproof performance and increase thrust torque, further improving operational efficiency. This pure electric propulsion system achieves the advantages of zero carbon emissions and environmental protection, minimizing the impact of boat movement on the ecosystem. The overall design is shown in Figure 2.

2.Autonomous navigation system:

This system is based on the Holybro Pixhawk 6C flight controller (Holybro, SDMODEL, Taiwan), Skydroid T12 remote control unit (Holybro, SDMODEL, Taiwan), and automatic identification system (AIS), complemented by the RFD900A high-power 2.4 G wireless data transmission system (IR-LOCK, Georgia, USA). Accurate positioning and reliable data transmission are critical issues in autonomous navigation systems. To address this, this system employs the HolyBro M9N GPS module, which supports the latest Global Navigation Satellite System (GNSS) [15]. This module offers rapid cold-start capabilities and high sensitivity, significantly reducing the positioning time and improving accuracy [16]. With this configuration, the USV can accurately anchor at the target location during water quality monitoring. The hardware equipment of the autonomous navigation system is shown in Figure 3.

To ensure safe navigation between the USV and surrounding vessels, the system is equipped with AIS. AIS enables the real-time identification of nearby vessels and provides critical information for collision risk assessments during navigation. Vessels equipped with AIS can broadcast their own status to be visible to others and receive information from surrounding AIS-enabled ships. Through publicly available international vessel tracking websites, it is possible to monitor ships currently transmitting AIS data, as shown in Figure 4 [17]. These platforms provide key navigational information such as the vessel name, heading, speed, and geographic coordinates, which can assist in route planning and enhance the navigational safety for the USV.

This system uses the RFD900A system for data transmission, which operates in the 902–928 MHz frequency band. It has strong sign penetration capabilities, effectively preventing the signal from encountering obstacles such as trees or buildings during navigation. Additionally, the integration of the Skydroid T12 remote control system, which utilizes Frequency Hopping Spread Spectrum (FHSS) [18], further reduces the risk of signal loss (data, video, and control) due to distance or environmental interference.

The USV supports both automatic and manual control modes. In the autonomous mode, positioning, route and water quality testing planning, and video transmission are managed through Mission Planner and Skydroid ground control station, as shown in Figure 5. The process flow chart is shown in Figure 6. The operators can use the T12 remote controller in manual mode to send control commands. These commands precisely adjust the output signal of the Pixhawk 6C servo system, enabling the real-time operation of the high-power underwater thruster for flexible control of the USV as needed. In addition to GPS and AIS, the USV is equipped with a mini camera in front of the integrated control box. This camera provides real-time video during manual control, which assists the operator in identifying obstacles such as floating debris or branches.

3.Water quality detection system:

This system integrates the pH sensor, DO sensor, ORP sensor, EC sensor, Pixhawk 6C, and the data acquisition subsystem through a microcontroller unit. The selection of sensors in this system was based on the ability to effectively reflect critical water quality characteristics in both river and port environments.

pH: this sensor measures the acidity or alkalinity of the water, which directly affects the living environment of aquatic organisms, metal ion solubility, and the chemical stability of various compounds in marine environments. It is susceptible to influences such as wastewater discharge, biological activity, and CO_2_ fluctuations, making it a reliable indicator of natural and anthropogenic water chemistry changes.DO: this sensor measures the oxygen concentration available for aquatic organisms. Low DO can represent organic pollution, excessive microbial respiration, or stagnation, while high levels may result from intense photosynthetic activity, such as during algal blooms. Tracking DO over time helps to evaluate the ecological health and oxygen dynamics of the water body.ORP: this sensor measures the overall redox condition of the water, which reflects the balance between oxidizing and reducing substances. This parameter helps identify biologically active zones, detect the presence of decaying organic matter, and infer the degree of water pollution. It often complements DO measurements, offering a more comprehensive view of the aquatic chemical environment.EC: this sensor measures the water’s ability to conduct electricity, which is directly related to the concentration of dissolved ions such as salts and minerals. EC is particularly important in port and coastal areas where salinity levels vary, and it can also indicate possible contamination from runoff, industrial effluents, or sewage.

By analyzing variations in indicators such as pH, DO, ORP, and EC, it is possible to determine whether the river is polluted and identify the nature of the pollution (e.g., organic contamination or heavy metal contamination). These parameters also help assess whether the river is suitable for ecological environments, drinking, irrigation, or other purposes [19,20,21]. These indicators form the foundation of water quality assessments and provide reliable data for water management and environmental protection. However, considering that water quality sensors may be submerged in water for extended periods, they could become covered with algae, dirt, or sediment, which may reduce the measurement accuracy and even damage the equipment [22]. To address this, we designed a dedicated sensor lifting mechanism that secures each water quality sensor within an anti-fouling cover and connects it to an electric cylinder, as shown in Figure 7. The microcontroller can read the changes in the output frequency from Pixhawk 6C to determine whether to activate the lifting mechanism for water quality measurements. This design can effectively prevent the sensors from being submerged in water for a long time and prevent them from becoming stuck due to obstacles such as branches when they are suspended and not in use. After each measurement, the collected water quality data are stored on an SD card. A detailed flowchart of the water quality detection process is shown in Figure 8.

The integrated USV system combines three core subsystems to achieve efficient and autonomous river water quality monitoring. The pure electric propulsion system provides an environmentally friendly and stable power source, ensuring minimal noise, reduced vibration, and zero emissions. The autonomous navigation system, equipped with high-precision GNSS modules and long-range wireless communication, enables real-time remote control and precise positioning. At the same time, the water quality detection system provides accurate water quality data for environmental detection, which form the basis for environmental assessment and pollution classification. Figure 9 and Figure 10 illustrate the overall structural configuration and functional flowchart of the USV. These diagrams demonstrate the coordinated operation among all modules, including data acquisition, control commands, and propulsion work in synchronization. When the USV reaches the designated location, the flight controller stops the propulsion system by disabling the PWM output to the thruster, effectively halting the USV’s movement. Once the vessel is stationary, the flight controller sends a separate set of PWM to the microcontroller. Upon receiving this signal, the microcontroller activates the electric cylinder to initiate the water quality detection mission. This integration reduces manual intervention and provides an intelligent platform for future environmental monitoring applications, offering innovative support for river water management and ecological conservation.

## 3. Results Demonstration

Before system implementation and field deployment, it is important to recognize that water quality conditions significantly differ between river and port environments. River water is typically influenced by inland runoff, organic pollution, and variable flow velocities, which cause fluctuations in parameters such as pH, DO, and ORP. In contrast, port and coastal waters are more strongly affected by salinity intrusion and industrial marine activity, leading to a relatively high EC, a more alkaline pH, and potential contamination by petroleum compounds or heavy metals.

According to the Ministry of Environment (Taiwan), typical acceptable ranges for river and coastal water quality are as follows:pH: 6.5~9DO: ≧6.5 mg/LORP: 200~400 mVEC: the EC of river water is about 50~1000 μS/cm, while that of port or coastal waters is about 30,000~55,000 μS/cm.

These baseline differences influence the expected sensor readings and the challenges of measurement accuracy, sensor fouling, and USV maneuverability in each environment. Therefore, we conducted two field trials in different aquatic settings—the Love River and the port of Kaohsiung (Cijin)—to validate the adaptability and robustness of the developed USV across variable environmental conditions.

The Love River is a narrow inland water body with limited ship activity. The trial focused on evaluating waypoint-based autonomous navigation and water quality detection performance under low-interference conditions. The port of Kaohsiung (Cijin) is a complex and busy maritime environment. The trial was designed to validate the USV’s ability to perform autonomous navigation and avoid potential risks in the presence of high salinity, fluctuating currents, and active surrounding ship movements.

### 3.1. Field Trial: USV Operation and Water Quality Monitoring

To validate the proposed USV, an initial field sailing trial was conducted at the Love River in Kaohsiung City, Taiwan, as shown in Figure 11. The experiment was divided into two stages: waypoint planning and navigation in automatic mode and remote operation with real-time video assistance in manual mode. In the automatic mode, after setting waypoints through Mission Planner, each designated location was marked in green on the interface. When the waypoint planning was complete, the USV autonomously navigated along the planned route and automatically conducted water quality detection around the designated locations without human intervention, as shown in Figure 12. Due to the lack of ship activity around this navigation, no AIS signals could be received, so no identification information of any nearby ships was displayed in the image. In manual mode, the operator used real-time video feedback to monitor the area in front of the USV, allowing for timely headings adjustment and avoiding collisions with floating objects that could cause the boat to capsize, as shown in Figure 13. The field trial demonstrated that the water quality detection system could be effectively operated in both modes and successfully complete the measurement tasks, as shown in Figure 14. During the trial, the GPS module successfully received signals from 27 satellites, and the horizontal dilution of precision (HDOP) reached values as low as 0.6 [23], indicating high positioning accuracy. Moreover, the water quality data collected on the SD card closely matched the official monthly water quality reports published by the Kaohsiung city government. A comparison of water quality data collected by the USV and data provided by the government for public review is shown in Figure 15. At each designated measurement point, the USV remains for 1 min to conduct water quality measurements. The sensor was placed in the water for the first 20 s, and samples were collected every 10 s thereafter. The measured parameters were all within the range of the government-reported values, demonstrating the accuracy and practical applicability of this system.

### 3.2. Second Field Trial: Performance Validation of Autonomous Navigation

The second field trial was conducted in the port of Kaohsiung, Cijin District, Kaohsiung City. The Cijin sampling site exhibited higher pH and EC values than the Love River. This can be attributed to the higher salinity of seawater, which increases both the electrical conductivity and the alkalinity of the water. Conversely, because salinity affects the solubility of DO in water, the DO of seawater is lower than that of a river. From Table 2, we can see that the DO in Love River is relatively low. This suggests that the low DO in the river is more likely attributed to organic pollution or eutrophication. To enhance the visibility of the USV during the sailing trial, the hull was repainted in high-contrast colors before deployment. This adjustment allowed for the easier identification of the USV’s position in recorded images, as shown in Figure 16.

Unlike the first trial, which primarily focused on linear navigation along the Love River, the second trial was specifically designed to evaluate the USV’s autonomous navigation capabilities in a more complex environment. In this trial, there were many ships moving and docking around the trial site. Before the trial began, the dynamics of the surrounding ships could be clearly understood through the data sent back by the AIS, as shown in Figure 17. The USV was required to follow a multi-point circular path around the part of Kaohsiung, simulating real-world patrol or monitoring scenarios. Users configured the mission by setting a series of waypoints through the control interface. During execution, the real-time monitoring window (left panel) displayed key status information such as GPS signal strength and battery level, allowing operators to assess the USV’s condition and make informed decisions about mission continuation, as shown in Figure 18.

The collected water quality data are shown in Figure 19. The results closely match the official water quality reports for Cijin by the Kaohsiung city government, further confirming that the proposed USV-based monitoring system maintains high accuracy and reliability even under different environmental conditions. These field trials also revealed several key challenges caused by differences in the surrounding environment and water conditions. In the port area, high salinity and suspended particles increased the risk of sensor fouling, which may reduce measurement accuracy. The stronger currents and wave disturbances affected the USV’s maneuverability and required the careful tuning of propulsion settings.

## 4. Conclusions

In this research, we successfully developed a USV for water quality detection with dual control modes—automatic and manual. The field trial results show that the water quality detection system operates stably and successfully collects water quality data in both modes. The measurement results, such as pH, DO, ORP, and EC, effectively identify the presence of industrial wastewater or chemical discharges in the river, fully demonstrating the reliability and stability of the USV. Unlike conventional remote-controlled USVs, the USV developed in this study supports autonomous navigation. With the integration of AIS, the USV can also identify and avoid nearby vessels during navigation. In addition, the integration of the electric cylinder and the sensor lifting mechanism allows the USV to perform water quality detection when arriving at designated sampling points. This design prevents the sensors from being submerged in water for a long time, reducing the risk of sensor degradation and ensuring long-term measurement accuracy.

In the future, the USV will further optimize the accuracy of autonomous navigation in automatic mode. Currently, when the USV encounters obstacles, it still relies on manual observation and control to adjust the original route, and it cannot autonomously avoid obstacles. To address this, future iterations will add sensors, such as radar and infrared, to enhance the USV’s environmental awareness and autonomous obstacle avoidance. AIS information will be incorporated into the navigation logic to support decision making during operation. These improvements aim to enable the system to perform autonomous obstacle avoidance by increasing its awareness of surrounding conditions. Additionally, the system currently depends on commercial charging infrastructure to power its battery. To improve energy efficiency and operational endurance, solar panels may be integrated into the USV’s upper surface. This would support passive battery charging and help reduce the internal temperature under prolonged sunlight exposure, further extending battery life and enhancing system sustainability in long-term environmental monitoring applications.

## Figures and Tables

**Figure 1 sensors-25-03747-f001:**
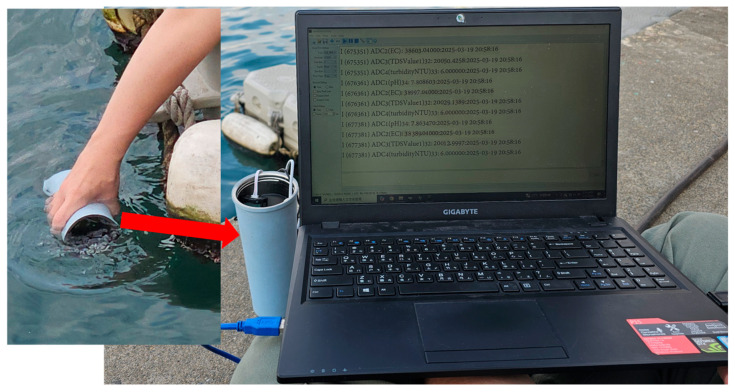
Manual sampling of water quality.

**Figure 2 sensors-25-03747-f002:**
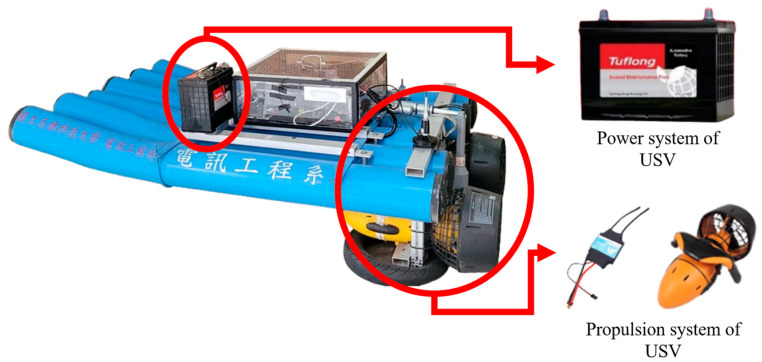
The design of the pure electric propulsion system on the USV.

**Figure 3 sensors-25-03747-f003:**
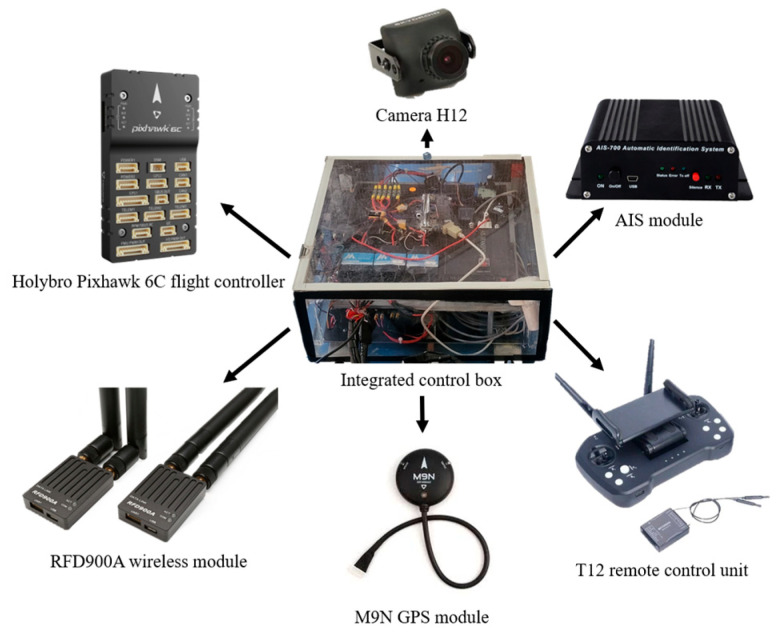
Hardware equipment of the autonomous navigation system.

**Figure 4 sensors-25-03747-f004:**
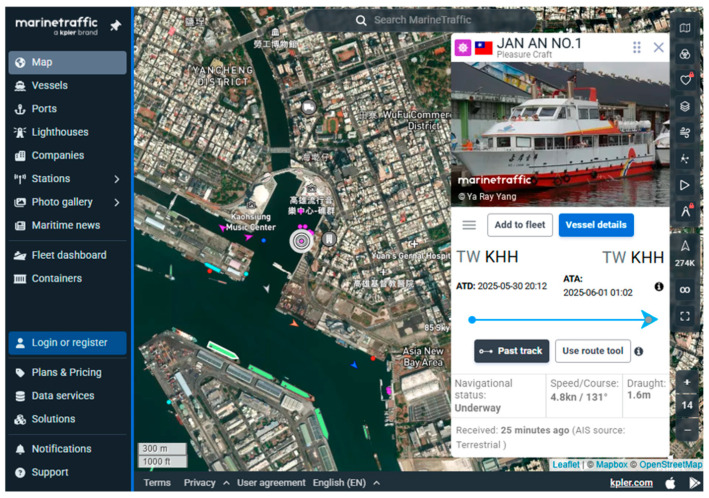
International vessel tracking website.

**Figure 5 sensors-25-03747-f005:**
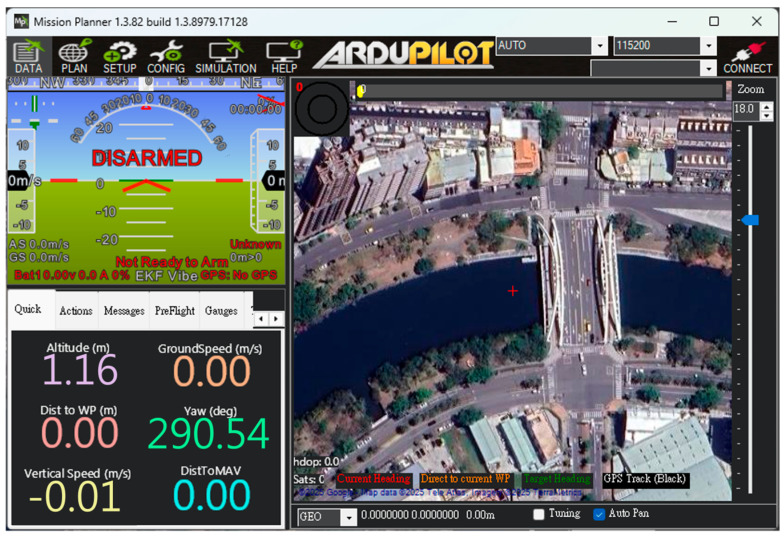
The control interface of Mission Planner.

**Figure 6 sensors-25-03747-f006:**
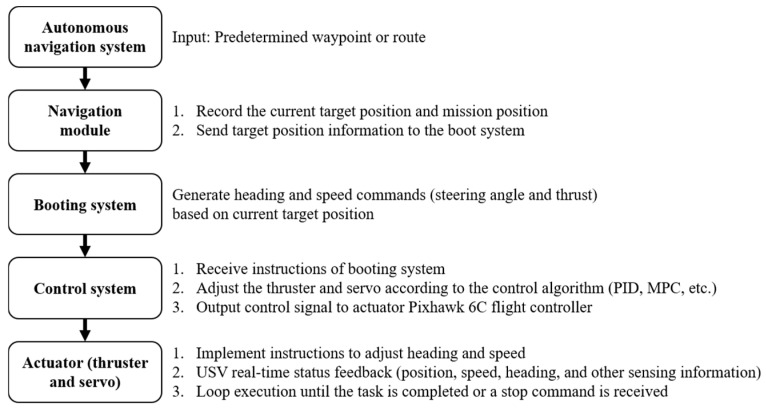
Flow chart of autonomous navigation.

**Figure 7 sensors-25-03747-f007:**
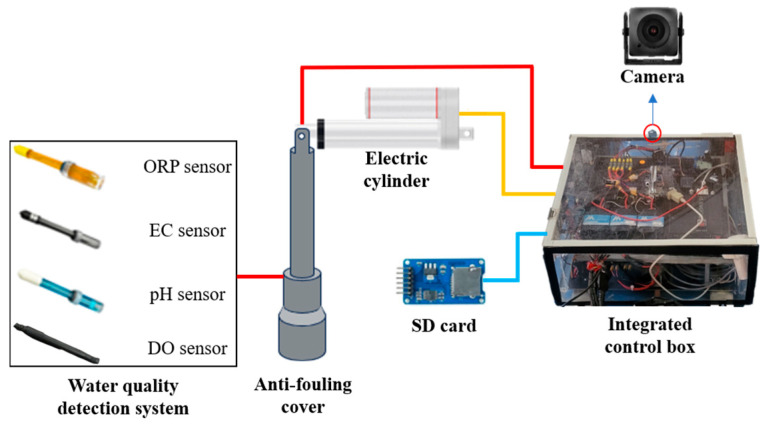
The structure of the water quality detection system.

**Figure 8 sensors-25-03747-f008:**
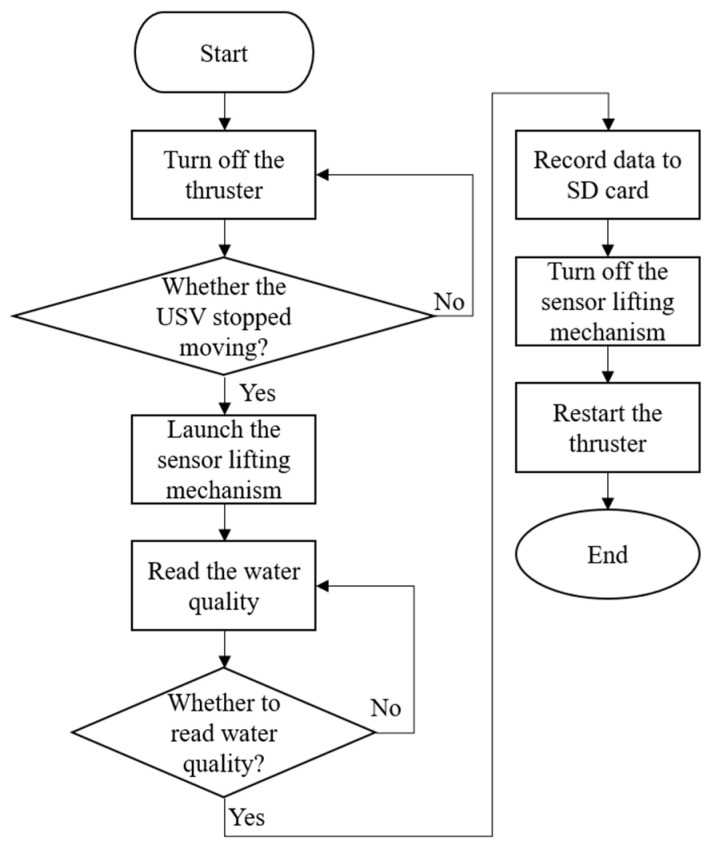
Flowchart of the water quality detection system.

**Figure 9 sensors-25-03747-f009:**
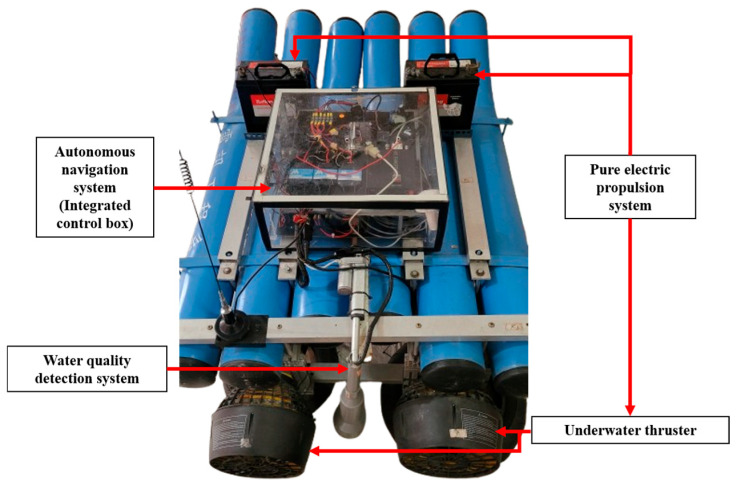
The structure of the USV.

**Figure 10 sensors-25-03747-f010:**
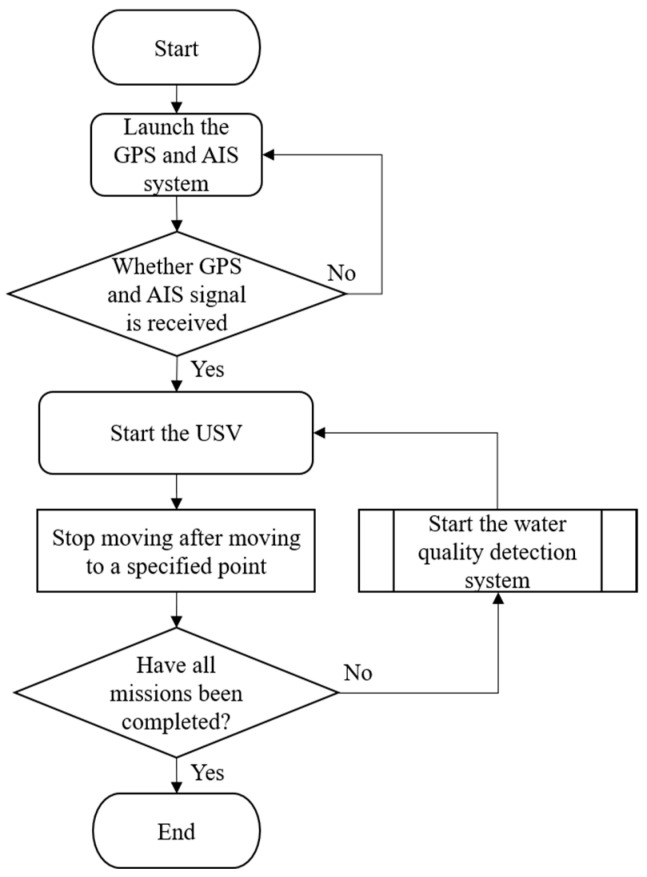
Flowchart of the navigation and water quality detection.

**Figure 11 sensors-25-03747-f011:**
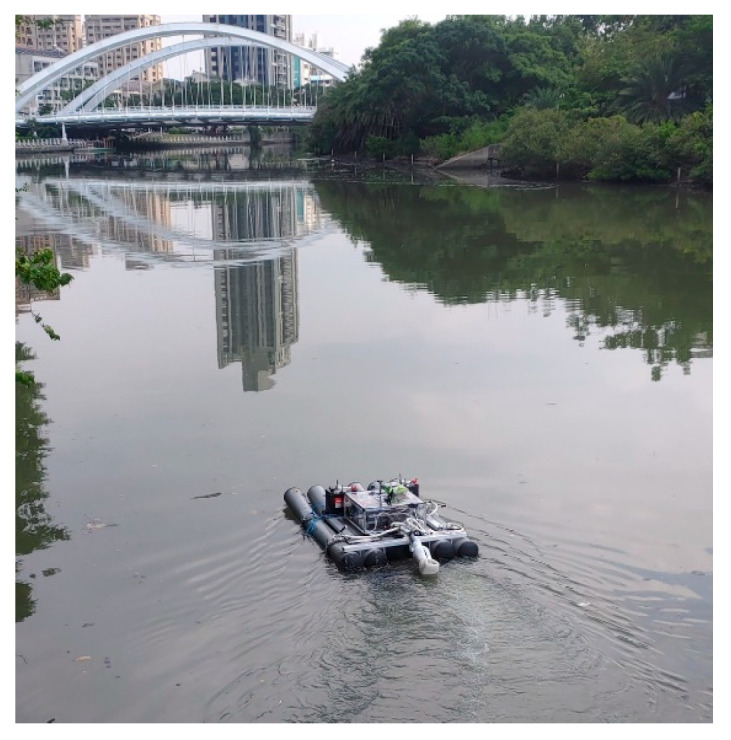
The sailing trial at Love River.

**Figure 12 sensors-25-03747-f012:**
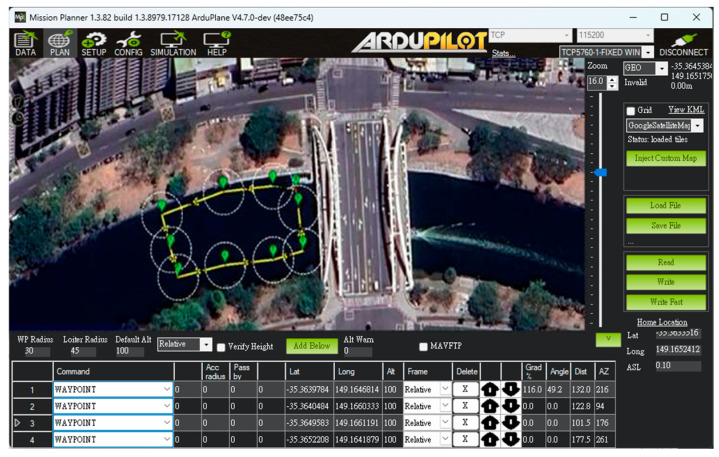
Waypoints setting by Mission Planner.

**Figure 13 sensors-25-03747-f013:**
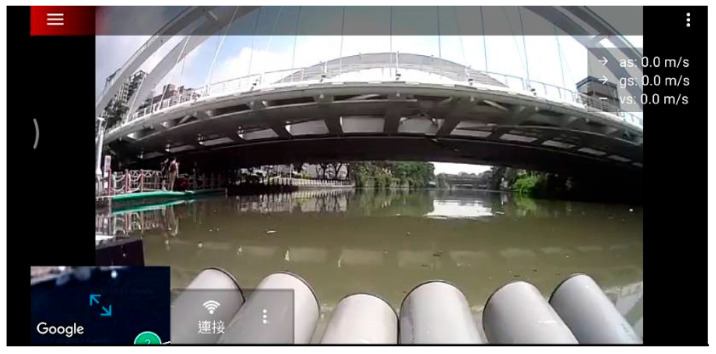
Feedback screen of the USV.

**Figure 14 sensors-25-03747-f014:**
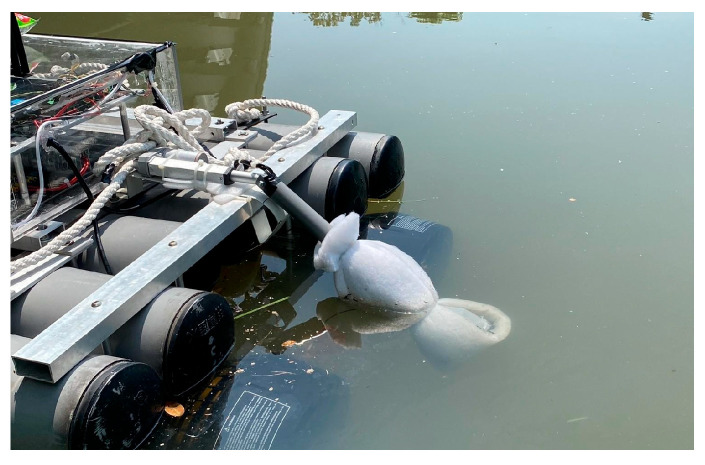
Process of conducting water quality testing.

**Figure 15 sensors-25-03747-f015:**
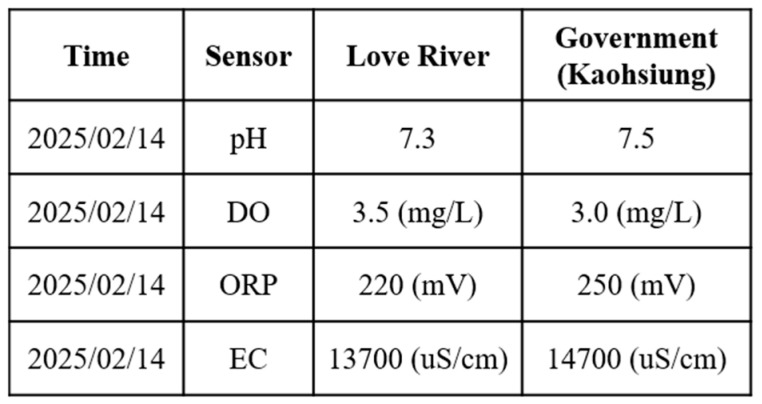
Comparison of water quality detected by the USV and the government at Love River.

**Figure 16 sensors-25-03747-f016:**
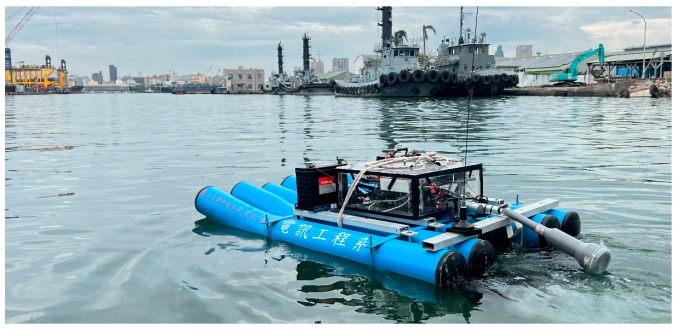
The USV is painted blue and sails on the sea.

**Figure 17 sensors-25-03747-f017:**
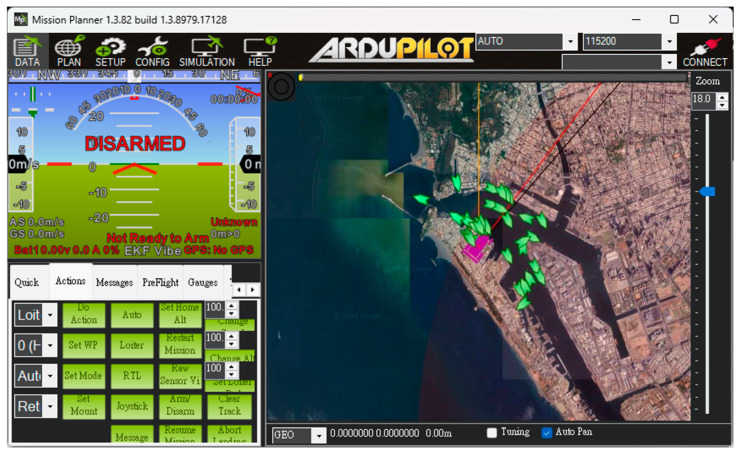
The dynamic of the surrounding ships on Mission Planner.

**Figure 18 sensors-25-03747-f018:**
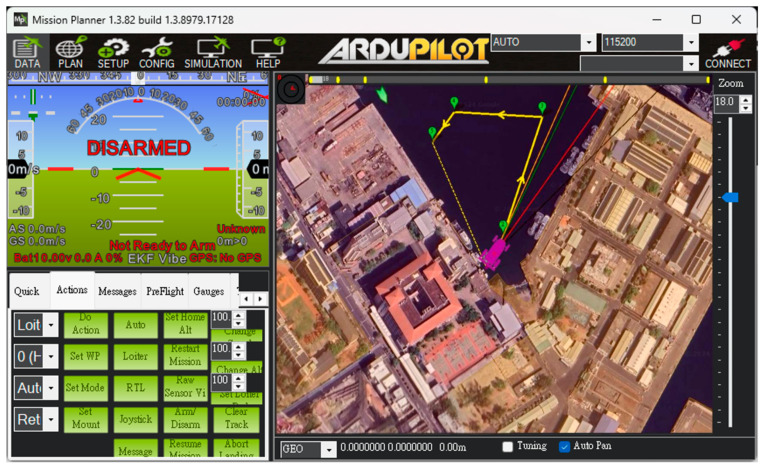
Waypoint setting at the port of Kaohsiung.

**Figure 19 sensors-25-03747-f019:**
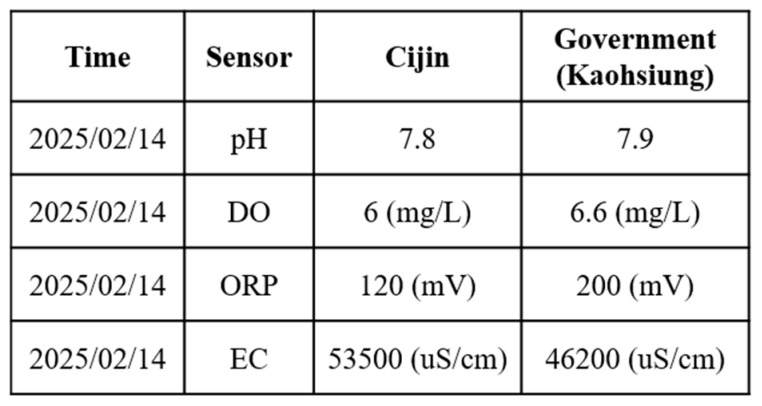
Comparison of water quality data from the USV and the government at the port of Kaohsiung.

**Table 1 sensors-25-03747-t001:** The weight value of water quality parameters.

Parameter	Weight Value	Unit
DO	0.24	Saturation
BOD	0.18	mg/L
pH	0.13	-
Ammonia nitrogen	0.15	mg/L
Coliform	0.12	Log (MPN/100 mL)
Suspended solids	0.11	mg/L
Total phosphate	0.17	mg/L (as P)

**Table 2 sensors-25-03747-t002:** Comparison of water quality of Cijin and Love River.

Parameter (25 °C)	Love River	Cijin
pH	7.3	7.8
DO	3.5 mg/L	6 mg/L
ORP	220 mV	120 mV
EC	13,700 μS/cm	53,500 μS/cm

## Data Availability

Not applicable.

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
