# Peer review of "Development of Autonomous Electric USV for Water Quality Detection"

_sensors, 2025, doi:10.3390/s25123747_

Round 1
Reviewer 1 Report (Previous Reviewer 2)
Comments and Suggestions for Authors
This paper is acceptable.
Author Response
The reply to the reviewer is as attached.

Reviewer 2 Report (New Reviewer)
Comments and Suggestions for Authors
In this paper, aiming at water quality detection, an electric unmanned surface vehicle (USV) was developed. The system structure and design, including pure electric propulsion system, autonomous navigation system, water quality detection system were introduced. The field trials and results were given in details. I think the research is novel and interesting. It has merit to be accepted in this journal. A few suggestions are listed as follows.
- It is recommended to give more certain data and information in the abstract, such as the distinctive and innovative features about the works. The USV for water quality detection is an engineering and commercial product, what the main advantages of this work compared with other USVs?
- In the paper, the author gave two field trials and results, what the differences between the field trials? What the main challenges to USV caused by various surrounding environment and sea/water conditions?
- How many samples do the USV collect in order to get the results in Fig.13 and 17?
- The size of the characters in different figures and tables are non-uniform. Please reorganize them according to the form requirements of this journal.
Author Response
The reply to the reviewer is as attached.

Reviewer 3 Report (New Reviewer)
Comments and Suggestions for Authors
This paper design an electric USV for remote river water quality monitoring. The article provides a good introduction to the USV system, the process and results of water quality detection. The point-by-point comments are given as follows:
- The author did not provide a detailed introduction to the autonomous navigation system, especially the display of the equipment. The reviewer suggests that, like the other two subsystems, a structural diagram should be provided.
- Regarding the startup mechanism of the water quality detection system, how to ensure that the detection system does not work, when the entire USV system is powered on but the USV is not started.
- As the core subsystem of the USV system, navigation guidance and control, is only described in detail in the manuscript for the autonomous navigation system. When the USV has an automatic mode, an introduction to the guidance and control system should be added.
- The reviewer suggests replacing the word ‘Ministry’ on line 209 with ‘Department’.
- The manuscript does not mention the camera sensor. How was Figure 11 obtained?
Round 2
Reviewer 2 Report (New Reviewer)
Comments and Suggestions for Authors
Thank you for the authors’ efforts,the revision has notably improved the paper,I have no further questions.
Author Response
Please see the attachment.

Reviewer 3 Report (New Reviewer)
Comments and Suggestions for Authors
- This article does not explain the difference between itself and other USVs for water quality detection, in order to highlight the contribution of the work.
- The author did not display the hardware equipment of the autonomous navigation system as requested by the reviewer. From the existing figures in the article, the reviewer did not see the camera.
- The starting mechanism of the water quality detection system is unclear, and it is not simply stated that it can be started when the ship is not in motion.
Author Response
Please see the attachment.

This manuscript is a resubmission of an earlier submission. The following is a list of the peer review reports and author responses from that submission.
Round 1
Reviewer 1 Report
Comments and Suggestions for Authors
This paper proposed an Unmanned Surface Vehicle (USV) for river water quality detection. The water quality detection system includes sensors for pH, dissolved oxygen (DO), electrical conductivity (EC), and oxidation-reduction potential (ORP). However, the paper lacks significant innovation. The use of USVs for water quality detection has already been reported in related literature, and the authors should emphasize the distinct aspects of their research, such as water quality sensor design, hull design, control methods, etc. In the current paper, the authors seem to have merely integrated various sensors into the USV without highlighting any notable technical features. Additionally, the experimental data provided is relatively limited.
Reviewer 2 Report
Comments and Suggestions for Authors
This paper report a pure electronic unmanned surface vehicle for water quality detection. But, what are the difficulties with this system? There is no in-depth analysis in the whole article.